# Cell Autophagy in NASH and NASH-Related Hepatocellular Carcinoma

**DOI:** 10.3390/ijms23147734

**Published:** 2022-07-13

**Authors:** Utibe-Abasi S. Udoh, Pradeep Kumar Rajan, Yuto Nakafuku, Robert Finley, Juan Ramon Sanabria

**Affiliations:** 1Department of Surgery, Marshall University Joan C. Edwards School of Medicine, Huntington, WV 25701, USA; udohu@marshall.edu (U.-A.S.U.); rajan@marshall.edu (P.K.R.); nakafuku@live.marshall.edu (Y.N.); rob3cjfin@gmail.com (R.F.); 2Marshall Institute for Interdisciplinary Research, Marshall University Joan C. Edwards School of Medicine, Huntington, WV 25701, USA; 3Department of Nutrition and Metabolomic Core Facility, Case Western Reserve University School of Medicine, Cleveland, OH 44106, USA

**Keywords:** cell autophagy, NASH, NASH-related HCC

## Abstract

Autophagy, a cellular self-digestion process, involves the degradation of targeted cell components such as damaged organelles, unfolded proteins, and intracellular pathogens by lysosomes. It is a major quality control system of the cell and plays an important role in cell differentiation, survival, development, and homeostasis. Alterations in the cell autophagic machinery have been implicated in several disease conditions, including neurodegeneration, autoimmunity, cancer, infection, inflammatory diseases, and aging. In non-alcoholic fatty liver disease, including its inflammatory form, non-alcoholic steatohepatitis (NASH), a decrease in cell autophagic activity, has been implicated in the initial development and progression of steatosis to NASH and hepatocellular carcinoma (HCC). We present an overview of autophagy as it occurs in mammalian cells with an insight into the emerging understanding of the role of autophagy in NASH and NASH-related HCC.

## 1. Introduction

Non-alcoholic fatty liver disease (NAFLD) is a spectrum of disease that ranges from steatosis to its inflammatory form, non-alcoholic steatohepatitis (NASH), which can progress with or without advanced fibrosis (cirrhosis) to hepatocellular carcinoma (HCC) [1,2]. It is one of the leading causes of liver-related morbidity and mortality in the Western countries, mainly due to a prevalence of sedentary lifestyle and increased consumption of high fat diets [1,2]. The forecasted prevalence of NAFLD in 2030 is conservatively 100.9 million cases worldwide, with NASH’s prevalence increasing up to 27 million cases [3]. NASH and its sequelae HCC have been rising as a global health challenge since it is more common in obese patients, and it has been estimated that 1.2 billion people will be overweight or obese by 2030. NAFLD is a condition in which an excess of fat from the diet accumulates in the liver in an absence of excessive alcohol consumption. In its inflammatory form, NASH is considered the initial phase of liver inflammation, damage, and fibrosis. Clinically, NASH is considered when serum aminotransferases are elevated in an obese subject with metabolic syndrome (hypertension and insulin resistance), or with incidental findings on abdominal imaging [4]. Histologically, it mimics alcoholic liver disease, showing inflammatory infiltrates, focal necrosis, and fibrosis [5]. It is well documented that NASH can progress to HCC, the most common type of primary liver cancer [6,7].

Growing evidence shows that the molecular mechanism underlying the progression of NAFLD to NASH is multifactorial, including metabolomic, immunologic, genetic, and endocrine pathways in concert with changes in gut microbiota communities [8]. Metabolically, NASH comorbidities include hyperlipidemia, insulin resistant status, obesity, and hypertension [9,10,11]. The presence of increased CD68+ cells, such as macrophages and regulatory T cells in affected livers with NASH [12], infers an important role of the host immunity in the regulation of inflammatory pathways. In addition, the loss of liver macrophages and Kupffer cells at the early stage of NASH development with subsequent infiltrations of Ly-6C+ monocyte-derived macrophages [13] reinforces the role of an immune response in the development of NASH. Genetically, there seems to be a difference in susceptibility to NASH progression among ethnic groups. NASH-related cirrhosis is more prevalent in European American descents compared to African American descents [14,15], even though there is a disproportionately large number of underdiagnosed African Americans with NASH [15]. The role of microbiota community changes in NASH is still controversial. Nevertheless, significant changes observed in gut micro-communities are consistent, albeit the specific bacterial species vary among authors [16].

The “two hit” hypothesis is one of the most widely accepted theories for the progression of NASH, proposing steatosis and insulin resistance as the initiating factors that set the stage from metabolic oxidative stress [17]. In addition, we would like to propose autophagy as another cellular process that enhances NASH progression. Autophagy, also referred to as “self-eating”, is a process found across many species in the eukaryotes. It involves the delivery and degradation of cytoplasmic materials by lysosomes and plays a prominent role in cell survival, differentiation, development, and homeostasis by controlling cytoplasmic physiology through energy balance and the removal of misfolded proteins, damaged organelles, and lipid droplets [18,19,20]. In mammalian cells, autophagy can be divided into three main types depending on the method in which the cytoplasmic cargo is delivered to the lysosomes, namely macro-autophagy, micro-autophagy, and chaperone-mediated autophagy [18,20,21].

## 2. Macro-Autophagy (MA)

MA, the most studied and major catabolic process used by eukaryotic cells to renew unfunctional proteins or unneeded organelles, operates physiologically at a lower rate, but its function is enhanced under stressful conditions, such as nutrient or energy deprivation [21,22]. It increases substrates that are needed for biosynthesis or energy production for cell survival through the degradation of cytosolic materials [21,23]. The MA process starts with the formation of a double membrane-bound vacuole known as autophagosome, which delivers targeted cytoplasmic materials for degradation to the lysosomes [19,24]. The autophagosome is formed via a non-selective or selective bulk sequestration of cytoplasmic “cargo” (portions of organelles, protein aggregates, or lipid droplets) that is to be delivered for degradation [19,24]. Selective MA targets a particular organelle or substrate cargo for sequestration and degradation by the autophagosome, i.e., mitophagy when mitochondria are the targeted cargo, ER-phagy—the endoplasmic reticulum, pexophagy—the peroxisomes, lipophagy for lipids removal, and aggrephagy for aggregate proteins, while ribophagy targets the ribosomes, and xenophagy—the removal of microorganisms invading the cytoplasm [19,25]. In selective MA, specific cargo is targeted to the autophagosomal membrane for sequestering via a ubiquitin-dependent or ubiquitin-independent pathway that involves autophagy adaptors such as sequestosome 1 (SQSTM1/p62), non-race-specific disease resistance 1 (NDR1), calcium-binding and coiled-coil domain-containing protein 2 (NDP52/CALCOCO2), and optineurin [19,24,26]. On the contrary, non-selective MA involves the degradation of materials by the lysosome in a non-specific manner [27]. The process of MA cargo delivery to the lysosomes for degradation is achieved via the movement of the autophagosome along microtubules, the acquisition of acidic and degradative properties through its fusion with endosomal compartments in the cells, and subsequent fusion with the lysosomal membrane [19,28]. Furthermore, MA is mediated by the activity of proteins encoded by autophagy-related genes (ATGs) and class III phosphatidylinositol 3-kinase (hVps34) in a complex with Beclin 1 and autophagy-related 14 (ATG14L) to produce phosphatidylinositol 3-phosphate (PI3P) that is required for autophagosome formation. Nevertheless, the maturation of the autophagosome into an autolysosome depends on certain autophagy-related (ATG) proteins including ATG14L, as well as on the activity of Ras-associated binding (Rab) GTPases, tethering factors (homotypic fusion and protein sorting (HOPS) complex), adaptors such as pleckstrin homology domain-containing family M member 1 (PLEKHM1), and soluble N-ethylmaleimide sensitive factor attachment protein receptors (SNAREs) for the final execution of the membrane-fusion process [19,20,24]. A summary of the three main types of autophagic processes in mammalian cells culminating in the final degradation of cargos by the lysosomes is shown in Figure 1.

### 2.1. Macro-Autophagy (MA) Process

The mechanism of MA involves primarily three processes, namely, (i) autophagosome formation, (ii) fusion of the outer membrane of the autophagosome with the lysosomal membrane to form an autolysosome, and (iii) degradation of the autolysosome contents by lysosomal enzymes. The formation of autophagosome is the first step in the MA process comprising initiation and nucleation, elongation, and, finally, the closure and maturation of the autophagosome. An autophagosome is a double membrane vesicle that is formed via the nucleation of a unique membrane structure known as a phagophore. Upon phagophore formation, cytoplasmic constituents are sequestered into it and, subsequently, the phagophore elongates, bends its membrane, and closes to give rise to the spherical-oriented autophagosome [18,21]. While in the yeast, phagophore formation starts at a single perivacuolar location called the phagophore assembly site (PAS); in mammals, phagophore formation could be initiated at several cytoplasmic locations, such as the endoplasmic reticulum (ER)-associated structures, known as omegasomes [21,29,30,31,32]. The origin of the phagophore membrane is still unclear, but it is hypothesized to come from the plasma membrane, Golgi complex, and/or mitochondria [21,24]. Once the autophagosome closes and matures, it moves along microtubules and fuses with the lysosomal membrane, resulting in the delivery of the sequestered cargo into the lysosome and the formation of a hydrolytic structure known as an autolysosome [21,22,24,33]. As part of the fusion process, there is a degradation of the inner membrane of the autophagosome due to its exposure to the acidic lumen of the lysosomes as well as its hydrolytic enzymes [18,21]. Upon a successful formation of the autolysosome, the autophagic enclosed materials are degraded by lysosomal hydrolases. The resultant components, including amino acids, lipids, and carbohydrate moieties that are derived from the autophagic degradation, are exported back to the cytoplasm through transporters and permeases for recycling or energy production [23,24,33]. In addition, in mammalian cells, the convergence of the MA pathway to the endocytic pathway often occurs, with the autophagosomes fusing with early or late endosomes to form structures referred to as amphisomes. Subsequently, the amphisomes fuse with the lysosomes to give rise to autolysosomes, leading to the final degradation of the autophagic cargo [21,34,35].

### 2.2. Regulation of Autophagosome Formation

The formation of the autophagosome includes the processes of initiation, nucleation, and elongation/maturation. 

#### 2.2.1. Initiation

In the yeast, a protein complex comprising Atg1, Atg13, Atg17, Atg31, and Atg29 kinases controls its initiation and formation. Its mammalian counterpart is made up of either Unc-51-like autophagy-activating kinase 1/2 (ULK1 or ULK2, which is an Atg1 homolog from the Unc-51-like kinase family), ATG13 (the mammalian homolog of Atg13), and RB1-inducible coiled-coil 1/focal adhesion kinase family-interacting protein of 200 kDa (RB1CC1/FIP200), which appears to be the ortholog of Atg17 of yeast [18,21,36,37]. Associated with this complex in mammals is C12orf44/ATG101, which binding to ATG13 is necessary for the MA mechanism [21,38]. The ULK1/2-ATG13-RB1CC1 complex is a very stable complex and forms irrespective of the cell nutritional status. This complex integrates the incoming autophagy signals to initiate the biogenesis of the autophagosome [18,21,39].

#### 2.2.2. Nucleation

The nucleation and assembly of the initial phagophore membrane involves the recruitment of another complex, namely, the class III phosphatidylinositol 3-kinase (PtdIns3K) complex, to the location of autophagosome formation [18,21,37]. This complex is formed by the interaction of Beclin 1 (which is the mammalian ortholog of yeast Atg6), with class III phosphatidylinositol 3-kinase and other key subunits including the PtdIns3K regulatory protein kinase p150 (hVps15), Atg14L, or Beclin1-associated autophagy-related key regulator (Barkor) and UV irradiation resistance-associated gene (UVRAG) [18,21,37,40,41,42]. In mammals and yeast, the PtdIns3K complex generates Phosphatidylinositol 3-phosphate (PtdIns3P), which is critical for the MA process [21,37,43].

#### 2.2.3. Elongation and Maturation

The elongation/maturation of the phagophore is mediated by two ubiquitin-like modification systems, the Atg12-Atg5-Atg16L (Atg16-like protein) complex, and LC3 (microtubule-associated protein 1 light chain 3) conjugation system [18,37,44,45,46]. In the Atg12-Atg5-Atg16L conjugation complex, the ubiquitin-like protein Atg12 is covalently tagged to Atg5. This process is mediated by the E1-like ubiquitin activating enzyme Atg7 and E2-like ubiquitin conjugating enzyme Atg10. Thereafter, the Atg12-Atg5 conjugate interacts non-covalently with ATG16L to form the Atg12-Atg5-Atg16L tetramers, which act as a E3-like ubiquitin ligase, playing a role in the autophagosome membrane elongation by heightening LC3 lipidation [18,21,47,48,49,50,51]. In mammals, the ATG12-ATG5-ATG16L1 complex associates with the pre-autophagosomal membrane, but it dissociates following the complete formation of autophagosome [21,37]. The second ubiquitin-like conjugation reaction involves the conjugation of LC3 to PE (phosphatidylethanolamine) with the help of E1-like Atg7 and E2-like Atg3 to give rise to LC3-II [18,21,37,52]. LC3-II is then specifically targeted to the elongating membrane to drive the closure of the autophagosomal membrane, forming a matured autophagosome [18,53].

One of the principal upstream signals that control the MA process, including the initiation step, is the mechanistic target of rapamycin complex 1 (mTORC1) [18,21,37]. In the cell, the association of the mTORC1 with the autophagy induction complex (ULK1/2-ATG13-RB1CC1) is nutrient-dependent. The availability of energy-producing compounds promotes the association of mTORC1 with the complex, leading to the phosphorylation and subsequent inactivation of ULK1/2 and ATG13 kinases, which drives the inhibition of autophagy. On the contrary, during starvation or when cells are treated with rapamycin, mTORC1 dissociates from the induction complex, driving the dephosphorylation of the proteins that make up the induction complex, resulting in the induction of MA [18,21,37,54,55].

## 3. Micro-Autophagy (mA)

mA is the lysosomal degradation process involving the direct engulfment of cytoplasmic cargos, without the involvement of autophagosomes as its transport intermediates. In mA, the lysosomal/endosomal membrane invaginates and forms a bud, which sequesters cytoplasmic material. This invaginated bud pinches off as a micro-autophagic body into the lumen of the organelle to be degraded and recycled [56,57,58,59]. Compared to other forms of autophagy, its mechanisms are poorly known, partly due to the significant overlap of its core components with other forms of autophagy, as well as the difficulty in isolating its functions for targeted studies [56,57,58,59].

### 3.1. mA Process

This involves five major steps. First is the constitutive invagination of the membrane of the lysosome/endosome, leading to the creation of an autophagic tube, a process that is upregulated under cellular starvation [60]. The formation of autophagic tubes is an active process, requiring the use of adenosine triphosphate (ATP) by vacuolar ATPases, and it is mediated by dynamin related GTPase Vps1p [61,62]. Starvation-induced mA is regulated via Atg7-dependednt ubiquitin-like conjugation (Ublc) systems that mediate membrane tethering [63,64]. In yeast, the vacuolar transporter chaperone (VTC) complex plays a role in the autophagic tube formation by modulating the membrane protein distribution, as well as serving as a site for calmodulin activation, which orchestrates acting binding proteins for autophagic tube formation [65]. Vesicle formation involves the sorting of membrane constituents with high-lipid and low-protein composition at the top of the autophagic tube [60]. Then, the vesicle binds with enzymes, implicating a reverse of the endocytosis process, and expanding the cytoplasmic leaflet [62]. Such vesicles have tendency to pinch themselves off into the lysosomal/endosomal lumen from the autophagic tube, and this vesicle scission does not require the various machinery required for MA [66]. After scission from the autophagic tubes, the released vesicles are broken down by Atg15 and hydrolyses and their contents are recycled via the actions of Atg22 [67].

### 3.2. Regulation of mA Activity

mA is active under cellular stress by starvation or nitrogen deprivation, maintaining organellar size and adjusting the membrane lipid composition by the allocation and degradation of excess cellular components through recycling [68,69,70]. mA works in association with MA and chaperone-mediated autophagy [71], but whether its function is limited to a compensatory mechanism from other forms of autophagy remains to be determined. mA is active in cells, with augmentation of the rate of invagination possibly from its interaction with MA-related pathways [72]. As the invagination and budding takes away portions of lysosomal membrane and lipid components, it contributes to the regulation of lysosomal size [69], in addition to controlling lipid metabolism [73]. Thus, mA serves an important role in cellular housekeeping [73].

### 3.3. Types of mA

There are various classifications of mA due to the diversity in its processes [74]. General mA in both yeast and mammals does not require the core autophagy machinery, but instead uses the endosomal sorting complex required for transport (ESCRT) [74,75,76]. General mA can be selective (micro-ER-phagy, micro-nucleophagy) or non-selective [74]. The functional classifications are based on the site of mA (lysosome or endosome) or sequestration process of cellular components (membrane invagination or extension) [74,77]. Another point of view proposed differentiating mA into fission and fusion types. The *fission type* occurs with the invagination and fission of endosomal/lysosomal membranes, requiring ESCRTs and conferring selectivity through the binding of ubiquitylated cargo to ESCRTs [74,78]. The *fusion type* occurs with invagination and extensions that become sealed by vertex fusion using core autophagy machinery with selectivity conferred with specific receptors [74]. Both mechanisms of mA are involved in the maintenance of cellular homeostasis and have been linked to neurodegenerative diseases such as Alzheimer’s disease and Huntington’s disease [79,80], as well as lysosomal glycogen storage diseases, such as Pompe disease [81].

## 4. Chaperone-Mediated Autophagy (CMA)

CMA is a selective, receptor-mediated form of autophagy that contributes to lysosomal proteolysis pathways that regulate the turnover of soluble cytosolic proteins [82]. The timely degradation of certain cytosolic proteins is essential in various cellular functions, such as the cellular response to stress, the metabolism of glucose and lipids, and DNA repairs. Thus, CMA plays an important role in cellular quality control and the supplementation of energy to cells under prolonged cellular stress [83]. The CMA process is a selective process, where target substrates are recognized and guided by their degradation tag (chaperone proteins), making the translocation across the lysosomal membrane a regulated process [84]. Since they require a targeting motif in their amino acid sequence that binds to HSC70 (heat shock protein-70 kDa), not all cytosolic proteins can be a candidate for lysosomal degradation through this process [85]. CMA does not require the formation of vesicles for the translocation of the target proteins into lysosomes, and thus, the direct lysosomal translocation of cargos occurs in CMA [86].

### 4.1. CMA Process

The selectivity in CMA is maintained by a sequence of the substrate protein, the pentapeptide KFERQ motif that enables its targeting for lysosome degradation. This motif is present in about 30% of the cytosolic proteins [87,88]. As with most intracellular targeting motifs, the KFERQ sequence exists in its inactive form and its specificity depends on the charges on the sequence component, rather than its amino acid combination [89]. The CMA-targeting motif contains, on one side, a glutamine (Q), one negatively charged acidic residue (D or E), one positively charged basic residue (K or R), a hydrophobic residue (F, I, L or V), and, lastly, a fifth residue, which can be positively/negatively charged. In some proteins, the motif can be hidden inside a protein folding before unfolding with post-translational modification, and in others it can be present in the pre-post-translational state to be eliminated if the targeted protein is present in excess amounts [89]. Crucial to the mechanism of CMA is HSC70s, the cytosolic chaperone that recognizes the KFERQ motif and assists with the protein complex translocation across the lysosomal membranes [85]. HSC70 is also responsible for clathrin disassembly from coated vesicles, and the folding of unfolded cytosolic proteins. HSC70 typically binds to the hydrophobic region of a protein to help the folding of unfolded or misfolded proteins, but upon binding to the KFERQ motif, it promotes protein degradation through CMA [85]. Other chaperones that interact with HSC70, such as heat-shock protein-40 kDa (HSC40) increase the effectiveness of HSC70 by forming a complex, but HSC70 seems to be the only known chaperone that directly binds to the KFERQ-like motif [90]. Other co-chaperones that interact with HSC70 may either help with the targeting of a certain substrate or unfolding of the substrate before the translocation across lysosomal membranes [90]. HSC70 exists as lysosomal or luminal HSC70. At the membrane, luminal HSC70 plays a role in the unfolding of substrate proteins [91], as well as the recycling of receptor proteins from CMA translocation complex by facilitating their dissociation after the substrate is internalized into lysosome [92]. Lysosomal HSC70 is needed for the completion of the translocation of substrate into lysosomes, in which the substrate undergoes enzymatic degradation. Once the complex of the bound substrate and chaperone is transported to the lysosomal surface, it binds to the lysosomal membrane and begins its unfolding and subsequent translocation into the lysosomal lumen; this process is saturable [93]. This saturability comes from the required binding of lysosome-associated membrane protein-type 2A (LAMP2A) to the CMA substrates. LAMP2A is one of the three splice variants of the gene *lamp2*: LAMP2A, LAMP2B, and LAMP2C. They have the same luminal region with different transmembrane and cytosolic regions, but LAMP2A is the only one required for CMA [94,95]. While cytosolic HSC70 is abundant in the cytosol, the LAMP2A level on the lysosomal membrane is restricted, limiting the CMA process rate, and its blockage leads to highly specific CMA inhibition. LAMP2A synthesis is shown to be increased during cellular stress, such as mild oxidative stress and hypoxia [96,97,98,99]. Heat-shock protein 90 kDa (HSP90), also a known chaperone, also binds to the luminal side of the lysosomal membrane and helps to stabilize the integrity of the LAMP2A complex during the translocation process [92].

### 4.2. CMA Activity

CMA activity is regulated by several pathways, but mainly by the mechanistic target of the rapamycin complex 2-protein kinase B-PH domain leucine-rich repeat protein phosphatase (mTORC2-Akt-PHLPP1) axis. The regulation of the CMA translocation complex was shown to be dependent upon the phosphorylation of the lysosomal glial fibrillary acidic protein (GFAP) [100]; in its unphosphorylated form, GFAP binds to the cytosolic binding motif of LAMP2A to form part of the CMA translocation complex. While GFAP activity is upregulated by Akt and the lysosomal kinase target of rapamycin complex 2 (mTORC2), GPAP is downregulated at the lysosomal membrane by the PH-domain leucine-rich repeat protein phosphatase (PHLPP1). At the basal state, mTORC2 activates Akt [101], which then phosphorylates GFAP, resulting in the basal inhibition of CMA [102]. When CMA is activated, PHLPP1 is recruited to the lysosomal membrane to inhibit Akt from phosphorylating lysosomal GFAP, which then promotes LAMP2A complex formation to drive CMA [102]. Additionally, the regulation of CMA through Akt is also associated with the insulin-phosphoinositide-3-kinase-3-phosphoinositide-dependent protein kinase 1 (INS-PI3K-PDPK1) pathway that regulates Akt activity [103].

### 4.3. CMA Activity in Health and Disease

Chaperone-mediated autophagy activity is required in many cell processes to maintain normal homeostasis, and its decreased activity has been associated with the development of some pathological conditions. 

(*a*) *Nutritional starvation stress*. CMA is known to be triggered via nutritional starvation. The degradation of many enzymes involved in metabolism by CMA is significantly increased in the liver during fasting [104,105]. CMA induced by starvation provides cells with the retrieved amino acids to be used for protein synthesis and energy source generation. This controlled and selective retrieval of cellular fuels and maintenance of protein biosynthesis during such cellular stress reaches its maximal turnover rate in around 12 h, following the initiation of MA processes that precede it, and continues to be active until the starvation is resolved [104,105,106]. In metabolically demanding organs like the liver, MA lasts for close to 8 h following starvation, starting with proteolysis and then switching over to the more preferential lipids degradation. If the starvation lasts longer, CMA becomes the main pathway to replenish the amino acids [104,105,106]. Those newly provided amino acids contribute to both protein synthesis and gluconeogenesis. Interestingly, even though their functions overlap, MA and CMA processes have non-redundant functions in cellular regulation. In CMA-impaired livers, MA can compensate for the lack of CMA in clearing damaged proteins, but cannot fully mitigate the cellular damage caused by the inability to adequately address the genetic damage and change in metabolic flux [104]. Highlighting the importance of CMA in energy supplementation during starvation, both in vitro and in vivo models with impaired CMA exhibited reduced levels of ATP in nutrient-deprived states [104,107]. 

(*b*) *Metabolic pathway regulation.* CMA is also involved in hepatic glycolysis regulation by degrading the glycolytic enzymes that stop hepatic glycolysis, and the inhibition of CMA leads to energy deficiency in peripheral organs [104]. Previous studies on a CMA-defective mouse model showed that many of the enzymes involved in the tricarboxylic acid (TCA) cycle are degraded by CMA during a period of starvation [104]. Moreover, CMA is involved in lipolysis as lipid carriers and coating proteins such as perilipins 2 and 3 (PLIN2 and PLIN3), and lipogenesis enzymes have been shown to be substrates targeted by CMA [104]. It is worth noting that lipid droplet coat protein removal during starvation is triggered via the 5′-AMP-activated protein kinase (AMPK)-dependent phosphorylation of PLIN2 [108]. 

(*c*) *Regulation of transcription*. CMA also play a role in the control of transcription, mainly through its action on factor-κB (NF-κB), by degrading its inhibitor NF-κ-B inhibitor-α (IκBα) [109] and selective degradation of myocyte-specific enhancer factor 2A (MEF2A) and myocyte-specific enhancer factor 2D (MEF2D) in neurons [110]. 

(*d*) *Immune response regulation.* CMA is associated with immune response regulation, through the degradation of stimulator of interferon genes protein (STING), which is involved in innate immunity [111]. I additionally plays a role in the degradation of Itch (ubiquitin ligase), and as a regulator of calcineurin 1 (Rcan1, a factor that has been implicated in the pathophysiology of Alzheimer’s disease) [96]. 

(*e*) *Cell cycle control.* CMA initiates cell-cycle progression following DNA repair through the targeted degradation of hypoxia-inducible factor 1-alpha (HIF1α), a cell-cycle progression regulator [98], and serine/threonine-specific protein kinase 1, which is a cell-cycle checkpoint kinase that is integral in DNA repair [90]. 

(*f*) *Cell senescence.* There is an age-dependent decrease in activity of CMA in most cells and tissues in both rodents and humans, as the binding and translocation of substrate proteins by lysosomes are reduced with age in direct proportion to a decrease of LAMP2A [88,96,111,112]. The decrease in LAMP2A is not transcriptional or post-translational (from gene expression), but rather due to decreased CMA receptor stability [113]. Earlier studies have shown that the overexpression of liver LAMP2A in the transgenic aging mouse model improved the maintenance of cellular homeostasis [114], implicating CMA as a potential player in NASH progression and HCC development.

CMA has been associated with several neurodegenerative and lysosomal storage disorders [115]. In neurodegenerative diseases, such as Parkinson’s disease many of the associated proteins in their unmodified state can serve as CMA substrates; nevertheless, their pathogenic variants are unable to be targeted and transported to lysosomes to be degraded, and their accumulation not only produces cellular damage, but further slows CMA efficiency [116,117,118]. Lysosomal storage disorders, such as galactosialidosis and Tay–Sachs disease, are genetic diseases that have dysfunctional lysosomal enzymes, aggravating the protein degradation process that leads to substrate accumulation and cellular toxicity [115,116,117,118]. CMA activities in cancer cell lines have upregulated LAMP2A levels [107,119], enhancing cell survival against cellular stressors such as hypoxia and oxidative stress [119,120,121]. CMA may aid cancer proliferation by the degradation of the inhibitors of cell proliferation, such as Rho-related GTP binding protein RhoE (RND3] [122], and pro-apoptotic protein Bcl-2-binding component 3 (BBC3) [123]. CMA has also been implicated as a tumor suppressor [124,125] by maintaining genomic stability through improving DNA repair [90].

## 5. Autophagy in Liver Physiology

Prolonged starvation induces liver cell autophagy, implicating the activation of metabolic pathways to replenish needed nutrients through the degradation of substrates during liver malnutrition and/or parenchymal damage [126,127]. During periods of nutrient deprivation, autophagy acts as one of the drivers of gluconeogenesis, ketogenesis, and ß-oxidation in the liver. Previous studies have shown that autophagy is involved in the degradation of glycogen and the formation of autophagic vacuoles dependent on cell energy status [126,128], which correlated with stress and hepatocyte amino acid deprivation [126,129,130]. In neonates, autophagy mediates the restoration of plasma glucose levels during fasting by gluconeogenesis from amino acids [127,131]. A non-selective autophagy drives the process of proteolysis, making available amino acids for use in the process of gluconeogenesis [127,132]. In addition, the selective autophagy of triglycerides stored in lipid droplets (lipophagy) fuels the ß-oxidation cycle [127,133]. Lipophagy also regulates the rate of very-low-density lipoprotein (VLDL) assembly by the release of fatty acids and degradation of apolipoprotein B [127].

Autophagy may play a key role in preventing cell death in the liver. Following the induction of hepatocellular necrosis by dimethylnitrosamine (DMNA), an increase in the number and size of autophagic vacuoles was observed [126,134]. Induced liver cell autophagy by carbamazepine (CBZ) decreased the hepatic load of mutant alpha1-antitrypsin Z (ATZ), and the severity of the deficiency in a mouse model of AT-deficiency-associated liver disease [127,135]. Additionally, the preservation of liver autophagic activity resulted in the lower intracellular accumulation of damaged proteins, an improved ability to handle protein damage, and an overall improvement in hepatic function [114]. Liver autophagy activity balances diverse metabolic pathways, but also removes damaged organelles (e.g., mitochondria, endoplasmic reticulum, peroxisomes), protecting and repairing liver cells from injury, thereby playing a key role in hepatocyte homeostasis [126].

## 6. Autophagy in NASH and NASH-Related HCC

Cell autophagy activity is impaired in NAFLD, NASH, and NASH-related HCC. Obesity and a prolonged high-fat diet affect the cell autophagy machinery at various stages, including the blockade of autophagosome formation, the inhibition of autophagosome–lysosome fusion, and the disruption of lysosome physiology [136,137]. Sustained high-fat diets affect lipid cell autophagy activity with increased accumulation of lipids in the liver, downregulating lipid β-oxidation and ATP production [136,137]. In addition, lipid accumulation causes alterations in the autophagosome membrane, leading to a non-efficient fusion process between autophagosomes and lysosomes [106,136]. The knocking down of autophagy-related genes, or their pharmacological inhibition, leads to the accumulation of triglycerides as lipid droplets [106,137], lower free fatty-acid oxidation, and decreased VLDL vesicle secretion from the liver cells [137]. Emerging data show that autophagy is impaired in the NAFLD cell line, in animal models of NAFLD, and in liver specimens from NAFLD patients [137,138]. Inducing hepatic autophagy through the increased expression of liver specific Atg7 in ob/ob mice ameliorated metabolic stress and attenuated hepatic steatosis [137,139]. Furthermore, the upregulation of pro-autophagic transcription factors such as FoxO1 and transcription factor EB (TFEB) prevented steatosis [137,140,141]. Previous studies have also revealed that enhancing autophagy prevented cell death in hepatocytes that had been exposed to prolonged treatment with palmitic acid [137,138]. In addition, treatment with carbamazepine and rapamycin, which are autophagy-inducing drugs, leads to a reduction in liver steatosis and triglyceride levels in both the liver and blood [137,142]. Nevertheless, the specific role of autophagy in the progression of steatosis to NASH and NASH-HCC remains elusive [143].

The loss of beclin-1 (an autophagy gene) leads to HCC development [143,144,145]. The knocking out of beclin-1 in mice results in spontaneous HCC [143,144,145], and in human HCC, the loss of beclin-1 is a common feature that correlates with poor prognosis [143,146]. Rubicon (a suppressor of the late stage of autophagy) is increased in NAFLD and plays a key role in driving hepatocellular fat accumulation and apoptosis [147,148]. Furthermore, cell autophagy activity is suppressed in hepatocyte-specific TGF β-activated kinase 1 (TAK1) knockout mice [137,149]. TAK1 serves as a positive regulator of AMPK, and its deletion in these mice led to increased mTOR activity and the suppression of autophagy, followed by severe hepato-steatosis [137,149]. Interestingly, these animals (with a hepatocyte-specific knockout of Tak1 developed spontaneous hepatocarcinogenesis, with high levels of p62/SQSTM in the hepatocytes. Restored autophagy activity by rapamycin was associated with both the downregulation of mTOR and the attenuation of liver cancer development and growth [137,149]. The deletion of protein tyrosine phosphatase receptor type O (PTPRO) in mice (a tumor suppressor), resulted in severe autophagy impairment [137,150]. Additionally, immunohistochemical staining from human samples revealed that hepatic PTPRO was reduced in livers from subjects with NAFLD compared to subjects with normal livers, while p62/SQSTM1 was increased [137,150].

Impaired cell autophagy activity may induce inflammation due to its role in clearing damaged mitochondria. The release of reactive oxygen intermediates and mitochondrial DNA leads to the inhibition of inflammasomes and Toll-like receptor 9 [137,151,152]. In addition, cell autophagy is implied in the degradation of p62/SQSTM1, an activator of nuclear factor kappa B (NF-κB), enhancing the transcription of pro-inflammatory cytokines [137]. The inhibition of cell autophagy activity in NASH leads to the accumulation of p62/SQSTM1, an autophagic substrate hypothesized to be involved in the formation of the Mallory–Denk bodies (MDB). MDB are present in ballooned hepatocytes and are key markers for NASH diagnosis. The upregulation of cell autophagy by rapamycin results in MDB resolution in mice [136]. Furthermore, the impairment of autophagy in tumor cells following metabolic stress also results in the accumulation of p62/SQSTM1, leading to increased retainment of damaged mitochondria, heightening oxidative stress and DNA damage [148,153]. The ubiquitin-binding protein p62 is a multifunctional protein involved in the activation the NF-κB-signaling pathway, as well as in the transcription of genes encoding for antioxidant proteins and detoxification enzymes through the activation of the nuclear factor erythroid 2–related factor 2 (Nrf2) transcription factor [127,154,155]. Defective autophagy decreases cell energy, resulting in a low cell redox state by decreasing recycling damaged organelles, DNA, aggregated proteins, and pathogens to maintain energy balance, promoting an anaerobic inefficient metabolism [136,156]. Concomitantly, previous studies have shown a progressive upregulation of p62 in obese patients with steatosis and NASH in comparison to subjects with a normal liver, thereby suggesting a downregulation of autophagy in obese patients with hepatic steatosis and NASH [136,138].

A complete pathway of how cell autophagy activity influences NASH progression to ESLD and HCC remains to be determined [157]. Nevertheless, it appears that impaired cell autophagy activity affects early liver fat accumulation and mitochondrial function, leading to a progressive widening of metabolic disturbances that enhance cell senescence and cell apoptosis. Further metabolic disturbances enhance epigenetic changes and NASH progression, promoting an apoptotic switch with subsequent liver malignant genesis [158].

## Figures and Tables

**Figure 1 ijms-23-07734-f001:**
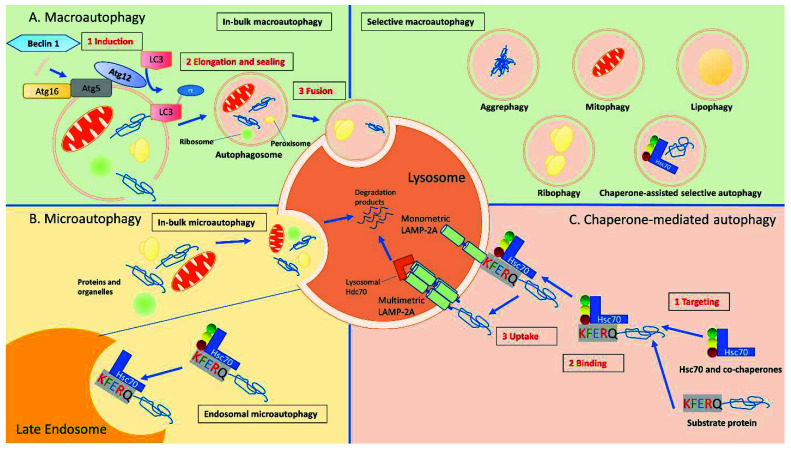
Autophagic pathways in mammalian cells. There are three major autophagic processes identified in most mammalian cells. (**A**) Macro-autophagy: in MA, cytosolic substrates determined to be transported are first packaged into autophagosomes, which are double-membrane vesicles formed through conjugation of autophagy-related proteins (i.e., Atg5, Atg12, Atg16) and autophagy-related protein LC3 with lipid PE. The formation of these autophagosomes is initiated by the phosphorylation of lipids in the membrane of organelles such as the endoplasmic reticulum, mitochondria, and Golgi apparatus, triggered by a kinase complex regulated by Beclin-1. Autophagosomes are targeted to lysosomes, and, after fusion of both vesicles, the cargo is delivered to the lysosomes for complete degradation. (**B**) Micro-autophagy: the in-bulk mA pathway allows cytosolic proteins and organelles to be degraded in bulk through invaginations at the lysosomal membrane. Through endosomal mA, cytosolic proteins can be selectively identified by Hsc70 to be transported to late endosomes using the KFERQ-like motif on the target proteins, leading to their internalization and degradation in the lysosomes. (**C**) Chaperone mediated autophagy: proteins in the cytosol with a KFERQ-like motif in their sequence can be identified by a molecular chaperone, Hsc70, and brought to the lysosomal membrane for translocation across the LAMP-2A multimeric complex. Lysosomal Hsc70 assist the translocation of the substrate protein, which is then degraded once inside the lysosomes (Adapted and modified from Scrivo et al. 2018) [20].

## Data Availability

No applicable.

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
