# Peer review of "Cell Autophagy in NASH and NASH-Related Hepatocellular Carcinoma"

_ijms, 2022, doi:10.3390/ijms23147734_

Round 1

Reviewer 1 Report

An interesting overview exploring a very timely topic in HCC management and landscape.

We would recommend some changes:

  • the background of HCC in the introduction is not comprehensively investigated. ICIs including pembrolizumab, nivolumab, durvalumab, atezolizumab, etc. have been recently evaluated in HCC patients, and clinical trials assessing single-agent ICI have reported disappointing results. Conversely, immune-based combinations have been more striking. In fact, the phase III IMbrave150 trial assessing the combination of the antiangiogenic agent bevacizumab plus the PD-L1 inhibitor atezolizumab versus single-agent sorafenib has established a new standard of care for HCC patients with advanced disease. Thus, the authors should include some recent papers discussing the current treatment scenario in HCC, especially considering that NASH is considered a possible predictive biomarker of response to immunotherapy in this setting (PMID: 34429006; PMID: 34431725 ; PMID: 35403533)
  • Minor linguistic mistakes and oversights to be corrected.
  • a more personal perspective section should be included, in order to better discuss how this topic could have a practical impact in the near future.

Some changes are necessary in my opinion. We recommend major revisions.

Author Response

(1) Background of HCC:

We have included  the background of NALFD, NASH and HCC in the introduction (see the revised version of the manuscript).

(2)  HCC treatment options:

Our paper is not focusing on current options in the treatment of HCC but on the possible role of autophagy in the pathogenesis of HCC, although we propose that targeting autophagy may be useful in the treatment of HCC.

(3)  Minor linguistic mistakes :

 We have checked and corrected all grammatical errors and mistakes.

Reviewer 2 Report

Journal: IJMS (ISSN 1422-0067)

Manuscript ID: ijms-1716003

Type: Review

Title: Cell Autophagy in NASH and NASH-related Hepatocellular Carcinoma

Section: Molecular Pathology, Diagnostics, and Therapeutics

Special Issue: Genetic and Molecular Mechanisms of Liver Disease

The manuscript is poorly organized and failed to focus on the title: Cell Autophagy in NASH and NASH-related Hepatocellular Carcinoma. Most text is about cell Autophagy, while Only section “Autophagy in NASH, and NASH related HCC” (lines 359-396) is about the main topic. My suggestion is to reject the manuscript. I have the following detailed suggestions and comments for the authors to improve the quality of manuscript.

Major comments

1. The title is “Cell Autophagy in NASH and NASH-related Hepatocellular Carcinoma”, but most text in the manuscript is not about NASH and NASH-related Hepatocellular Carcinoma. Only section “Autophagy in NASH, and NASH related HCC” (lines 359-396) is about the main topic. More text about the role of cell autophagy in NASH and NASH-related hepatocellular carcinoma should be added, and this section should be the main part of the manuscript. Please do a further search of related papers.

2. Please draw a figure to show the role of cell autophagy in NASH and NASH-related hepatocellular carcinoma.

3. I suggest changes of structures of the manuscript to as follows:

-1. Non-alcoholic steatohepatitis (NASH) and NASH-related hepatocellular carcinoma

-2. Cell Autophagy and mechanisms

-2.1 Macro-autophagy

-2.2 Micro-autophagy

-2.3 Chaperone-mediated autophagy

-3.The role of cell autophagy in NASH and NASH-related Hepatocellular Carcinoma (can be further divided to sub-sections, this is the main section of the manuscript)

-4. Future direction

-5. Conclusions

Minor comments

4. Abstract

Lines 37-40

“In non-alcoholic fatty liver disease (NAFLD), including its inflammatory form, Non-alcoholic steatohepatitis (NASH) a decrease in the autophagy cell activity is implicated in the initial development and progression of steatosis to NASH and hepatocellular carcinoma (HCC).”

Please check the English.

Author Response

  1. Additions of details on NASH/NASH-related HCC:
    We have added more details on molecular background of NAFLD/NASH/HCC in the introductory paragraph, as well as included a paragraph on the role of autophagy in NASH and HCC development.

  1. Figures for autophagy in NASH and NASH-related HCC:
    We believe we do not have sufficient information to draw a comprehensive figure without it being potentially misleading, as a lot of mechanisms and pathways are not well understood. Thus, we opted not to include a figure at this time.

  1. Manuscript structure change:
    We have included more details on the background of NAFLD/NASH/HCC and the relationship between autophagy and NASH/HCC, in line of what you have suggested .

4: Grammar:

We have revised and corrected grammatical errors and phrasings with contributions from multiple members of our lab.

Reviewer 3 Report

The reviwe is well written and organized. 

There are a few typographical errors or grammatical errors that should be fixed. Otherwise this reviewer does not have any substantial concerns.

Author Response

  1. Typographical errors:

We have revised and corrected grammatical errors and phrasings with contributions from multiple members of our lab. We have added sections on background of NAFLD/NASH/HCC, and the relationship between autophagy and NASH/HCC as another reviewer had suggested.

Reviewer 4 Report

This is a mini-review aimed to summarize the role of autophagy in NASH and NASH-HCC. A significant part of it is focus on general aspects of autophagy, being the part of the link between autophagy and NASH is really short. 

Author Response

  1. More details on autophagy and NASH:

We have added more details on molecular background of NAFLD/NASH/HCC in the introductory paragraph, as well as included a paragraph on the role of autophagy in NASH and HCC development. 

Round 2

Reviewer 1 Report

Acceptance.

Author Response

We thank you so very much for your comments

Reviewer 2 Report

Journal: IJMS (ISSN 1422-0067)

Manuscript ID: ijms-1716003-peer-review-v2

Type: Review

Title: Cell Autophagy in NASH and NASH-related Hepatocellular Carcinoma

Section: Molecular Pathology, Diagnostics, and Therapeutics

Special Issue: Genetic and Molecular Mechanisms of Liver Disease

The manuscript is poorly organized and failed to focus on the title: Cell Autophagy in NASH and NASH-related Hepatocellular Carcinoma.

The revised manuscript has improved, but not too much.

Most text is about general introduction of NASH and NASH related HCC or autophagy, while only section “Autophagy in NASH, and NASH related HCC” (lines 391-429) is about the main topic.

My suggestion is to reject the manuscript. I have the following detailed suggestions and comments for the authors to improve the quality of manuscript.

Major comments

1. The title is “Cell Autophagy in NASH and NASH-related Hepatocellular Carcinoma”, but most text is about general introduction of only NASH and NASH related HCC or only autophagy, while only section “Autophagy in NASH, and NASH related HCC” (lines 391-429) is about the main topic. More text about the role of cell autophagy in NASH and NASH-related hepatocellular carcinoma should be added, and this section should be the main part of the manuscript. Please do a further search of related papers.

I have made this comment last time, but the authors only added a small paragraph.

2. Please draw a figure to show the role of cell autophagy in NASH and NASH-related hepatocellular carcinoma.

I have made this comment last time, but the authors responded as follows: “We believe we do not have sufficient information to draw a comprehensive figure without it being potentially misleading, as a lot of mechanisms and pathways are not well understood. Thus, we opted not to include a figure at this time.”

If the information of role of cell autophagy in NASH and NASH-related hepatocellular carcinoma is not sufficient. Then a review of role of cell autophagy in NASH and NASH-related hepatocellular carcinoma is impossible and the design of the manuscript is sufficient. There are many review papers on only NASH and NASH related HCC or only autophagy. The manuscript like this one cannot be published in any scientific journals.

Author Response

We have added 2 pages on cell autophagy activity in liver physiology and NASH, and NASH related HCC. We have added the corresponding references. Although we thank you very much for your suggestion to improve our review, we in advance apologize for not complying with the inclusion of a figure, since the figure is part of an ongoing NIH grant close to be submitted as a publication based on our original data.

Reviewer 4 Report

This version of the manuscript is significantly better than the previous one. However, in order to be considered for publication I would include a section that connects the general description of autophagy with the section on autophagy and NASH/NASH-HCC. I would suggest to add a brief section (2-3 paragraphs) about the knowledge about the role of autophagy in liver physiology. Other ideas are also welcome.

Author Response

We have added 2 pages on cell autophagy activity in liver physiology and NASH, and NASH related HCC. We thank you very much for your suggestion

Round 3

Reviewer 2 Report

Journal: IJMS (ISSN 1422-0067)

Manuscript ID: ijms-1716003-peer-review-v3

Type: Review

Title: Cell Autophagy in NASH and NASH-related Hepatocellular Carcinoma

Section: Molecular Pathology, Diagnostics, and Therapeutics

Special Issue: Genetic and Molecular Mechanisms of Liver Disease

The manuscript is poorly organized and failed to focus on the title: Cell Autophagy in NASH and NASH-related Hepatocellular Carcinoma.

The revised manuscript has improved, but not too much.

Most text is about general introduction of NASH and NASH related HCC or autophagy, while only section “Autophagy in NASH, and NASH related HCC” (lines 427-494) is about the main topic.

My suggestion is to reject the manuscript. I have the following detailed suggestions and comments for the authors to improve the quality of manuscript.

Major comments

1. The title is “Cell Autophagy in NASH and NASH-related Hepatocellular Carcinoma”, but most text is about general introduction of only NASH and NASH related HCC or only autophagy, while only section “Autophagy in NASH, and NASH related HCC” (lines 427-494) is about the main topic. More text about the role of cell autophagy in NASH and NASH-related hepatocellular carcinoma should be added, and this section should be the main part of the manuscript.

I have made this comment last time, but the authors only did very minor revisions.

2. Please draw a figure to show the role of cell autophagy in NASH and NASH-related hepatocellular carcinoma.

I have made this comment last time, but the authors responded as follows: “We believe we do not have sufficient information to draw a comprehensive figure without it being potentially misleading, as a lot of mechanisms and pathways are not well understood. Thus, we opted not to include a figure at this time.”

If the information of role of cell autophagy in NASH and NASH-related hepatocellular carcinoma is not sufficient. Then a review of role of cell autophagy in NASH and NASH-related hepatocellular carcinoma is impossible and the design of the manuscript is sufficient. There are many review papers on only NASH and NASH related HCC or only autophagy. The manuscript like this one cannot be published in any scientific journals.

Author Response

We have done all the minor revisions as advised by the academic editor. Thank you

Reviewer 4 Report

The review has been much improved by the authors. I don't have further comments

Author Response

 Thank you